# Midpalatal Suture Maturation Stage in 10- to 25-Year-Olds Using Cone-Beam Computed Tomography—A Cross-Sectional Study

**DOI:** 10.3390/diagnostics13081449

**Published:** 2023-04-17

**Authors:** Anis Shayani, Marco Andrés Merino-Gerlach, Ivonne Angélica Garay-Carrasco, Pablo Eliseo Navarro-Cáceres, Héctor Paulo Sandoval-Vidal

**Affiliations:** 1Master Program in Dental Science, Faculty of Dentistry, Universidad de La Frontera, Temuco 4780000, Chile; 2School of Dentistry, Faculty of Medicine, Universidad Austral de Chile, Valdivia 5090000, Chile; 3Independent Researcher, Valdivia 5090000, Chile; 4Independent Researcher, Temuco 4780000, Chile; 5Centro de Investigación en Ciencias Odontológicas (CICO), Departamento de Odontología Integral de Adultos, Facultad de Odontología, Universidad de La Frontera, Temuco 4780000, Chile; 6Universidad Autónoma de Chile, Temuco 4780000, Chile; 7Department of Pediatric Dentistry and Orthodontics, Faculty of Dentistry, Universidad de La Frontera, Temuco 4780000, Chile

**Keywords:** midpalatal suture, midpalatal suture maturation stages, cone-beam computed tomography, maxillary expansion, ossification

## Abstract

In this study, we aimed to evaluate the frequency of midpalatal maturational stages in a Chilean urban sample of adolescents, post-adolescents and young adults, associated with chronological age and sex, by assessing CBCT scan images. Tomographic images in axial sections of the midpalatal sutures from 116 adolescents and young adults (61 females and 55 males, 10–25 years old) were classified according to their morphologic characteristics in five maturational stages (A, B, C, D and E), as proposed by Angelieri et al. The sample was divided into three groups: adolescents, post-adolescents and young adults. Three previously calibrated examiners (radiologist, orthodontist and general dentist) analyzed and classified the images. Stages A, B and C were considered to be an open midpalatal suture, and D and E were considered to be a partially or totally closed midpalatal suture. The most frequent stage of maturation was D (37.9%), followed by C (24%) and E (19.6%). The possibility of finding closed midpalatal sutures in individuals of 10 to 15 years was 58.4%, and in subjects aged 16 to 20 and 21 to 25 years, it was 51.7% and 61.7%, respectively. In males, Stages D and E were present in 45.4%; for females, this prevalence was 68.8%. Individual assessment of the midpalatal suture in each patient is of crucial importance before making the clinical decision of which is the best maxillary expansion method. Due to the extensive calibration and training required, it is advisable to always request a report from a radiologist. Individual evaluation with 3D imaging is recommended because of the great variability observed in the ossification of midpalatal sutures in adolescents, post-adolescents and young adults.

## 1. Introduction

Maxillary transverse deficiency may be one of the most common skeletal problems among orthodontic patients, causing posterior crossbite, dental crowding, occlusal disharmony, changes in tongue posture and mouth breathing, producing significant effects on muscle function and aesthetics [1,2,3].

The most effective orthopedic-orthodontic treatment for increasing the maxillary transverse width is maxillary expansion. The main concept related to this kind of treatment is to apply additional force to stimulate the separation of the midpalatal suture and the maxillary sutural system [4,5,6,7]. As treatment protocols, orthodontists suggest different types of procedures, such as conventional rapid maxillary expansion (RME) [8,9], miniscrew-assisted rapid palatal expansion (MARPE) [10] and surgically assisted maxillary expansion (SARME) [11].

RME has been routinely used to expand the transverse dimension and correct the transverse discrepancy. The therapeutic effect of RME manifests in skeletal and dentoalveolar effects. Skeletal expansion is the widening of the maxilla by opening the midpalatal suture, whereas dentoalveolar expansion involves dental tipping and bending of the alveolar process. RME is aimed at maximizing the skeletal effect and reducing dentoalveolar effect, which may be accompanied by side effects, such as loss of the periodontal attachment level, fenestration of the buccal cortical bone, root resorption and palatal tissue necrosis [12,13,14,15,16,17,18].

Closure of the craniofacial sutures and the midpalatal suture eventually make skeletal expansion by conventional RME impossible [6,19]. In these cases, more invasive treatments such as MARPE [20,21,22] or SARME [23] are preferred.

It is of great importance to determine the proper timing for palatal expansion, as its success is related to MPS fusion [24].

In orthodontics, several methods have been described for the evaluation of the skeletal age [25], including hand-wrist radiography [26,27] and cervical vertebral maturation (CVM) based on lateral cephalogram [28,29,30,31].

Radiation exposure to pediatric patients who are more radiosensitive than adults should be as low as possible. In this sense, it is very important to use reliable indicators for predicting midpalatal suture maturation, such as hand-wrist radiographs, with an effective dose of 0.16 μSv [32] for predicting the midpalatal suture maturation.

In the case of the assessment of skeletal age with the CVM, it is performed on a cephalometric radiograph, which is routinely used in orthodontic practice, which makes it easy to apply. The use of an X-ray protective thyroid collar is of crucial importance. The effective dose of a lateral cephalogram without a thyroid collar was shown to be higher than a hand-wrist radiograph at 5.03 μSv [32].

Even though both methods have proven very useful for the estimation of skeletal age, they do not allow for clinicians to observe the midpalatal suture in situ.

The start and the advance of the midpalatal suture vary greatly with age and sex. Different authors have reported patients at ages 27 [33], 32 [33], 54 [34] and even 71 [35] years having some signs of fusion of this suture. Such findings indicate that the variability in the developmental stages of fusion of the midpalatal suture is not directly related to chronologic age, particularly in young adults [33,34,35,36,37]. This was reaffirmed by Fishman [38], who mentioned a low correlation between skeletal and chronological age, evidencing the need for individual indicators of the patient’s skeletal and facial growth stages [39].

The closure of the MPS is influenced not only by the patient’s chronological age, but also by the great variability individuals present among themselves, such as nutritional deficiencies, the population’s ethnicity, the skeletal development itself and other structures that surround the suture [40,41,42].

Revelo and Fishman [43] proposed individual assessment of the midpalatal suture morphology with occlusal radiographs before RME therapy. Many authors mention that this method is unreliable, since the images of the vomer and external nose structures are superimposed on the suture area, leading to misinterpretation of the fusion stage of the palatal suture [37,39].

Computed tomography (CT) has emerged as an efficient tool for image diagnosis in situ [44], because it enables the assessment of skeletal maturation with a detailed registration of bone morphology. Consequently, the three-dimensional images obtained with CT may be essential for making clinical decisions that involve craniofacial development. Having a 3D diagnostic exam available is useful for the clinician when deciding the best treatment strategy for each patient, as it avoids the various radiographic investigations often necessary to obtain all the essential information needed to draw up an orthodontic treatment plan [45]. An important point to have in consideration is that it is impossible to regularly take CBCT from every pediatric patient, with an effective dose of radiation ranging between 19 and 368 μSv [32], due to ethical concerns about unnecessary radiation exposure. For this reason, each patient must be evaluated individually and in detail.

Methods based on CT have been developed for the assessment of maturation stages in the midpalatal suture [5,40,46]. The method of individual assessment proposed by Angelieri et al. [40] was designed to allow for a reliable and reproducible diagnostic pathway for clinical use [47].

The authors in [40] determined five maturational stages (A, B, C, D and E) as a way of providing more reliable clinical data when making the decision between RME or more invasive procedures. According to this, patients in stages A and B would have less resistance and greater skeletal effects of RME than those in stage C. For patients in stages D and E, MARPE-SARME was recommended [48].

The aim of this study was to evaluate the frequency of MPS stages in a Chilean urban sample of adolescents, post-adolescents and young adults associated with chronological age and sex, assessing MPS maturational stages using CBCT scans. This would justify requesting a CBCT for sutural diagnosis in patients of this age group, avoiding unnecessarily exposing patients to surgical procedures and the risks associated with them.

## 2. Materials and Methods

This article was prepared based on the Strengthening the Reporting of Observational studies in Epidemiology (STROBE) statement [49].

This descriptive, retrospective, cross-sectional study was approved by the Ethics Committee of the Universidad de La Frontera, Temuco, Chile, with the approval number 101/20. All patients signed a donation form.

The sample of this study included 116 patients, 61 females and 55 males, aged 10 to 25 years, retrospectively selected from the Dental Teaching Clinic at the Faculty of Dentistry at the Universidad de La Frontera, Chile. CBCT images were taken between June 2018 and June 2021. None of the patients underwent radiation exposure exclusively for the present study aim.

The sample size was determined through a sampling technique for finite populations, to which a sampling adjustment was made. It was performed with a 95% confidence level, with the precision of a 5% and 10% proportion of this possibility.

Inclusion criteria were age between 10 and 25 years and the availability of CBCT images. The exclusion criteria were a history of previous orthodontic treatment or any appliance at the examination (previous maxillary expansion as an early interceptive orthodontic phase may affect suture status), cleft lip and palate and syndromic conditions. The primary justifications for the CBCT request were the diagnosis of retained teeth (such as canine impaction), skeletal malocclusion, an assessment of third molars, evaluation of a discrepancy between the maxilla and mandible, dental inclination or thickness of bone tables.

The CBCT scan images were obtained with a Pax-i3D cone-beam imaging system (Vatech, Hwaesong, Republic of Korea).

For all scans, the minimum field of view used was 11 cm, 90 kV, 10 mA, and the scan time ranged from 8.9 to 20 s with a resolution of 0.25 to 0.30 mm.

Adjustment of the patient’s head in the three planes of space and the selection of a slice for evaluation of the MPS maturational stages were performed according to the protocol described previously [40].

Image analysis was performed using Ez3D Plus Software (Vatech, Hwaesong, Republic of Korea).

Based on the difficulties presented in some samples in obtaining complete visualization of the midpalatal suture in a single cut, it was decided to divide the maxilla into two cuts: the anterior region (palatine processes of the maxillary bone) (Figure 1) and the posterior region (palatine bone) (Figure 2). This small modification to Angelieri et al.’s method [40] allowed us to better standardize the systematic and meticulous observation of the suture.

The images were obtained as follows. First, in the multiplanar reconstruction screen, the skull image was manipulated so that the vertical and horizontal lines were overlaying the MPS in the axial and frontal cuts.

Afterward, in the sagittal view, the patient’s head was adjusted so that the horizontal reference line coincided with the median region of the palate, which is the cancellous bone between the upper and lower cortical bones.

After that, the final image was used in the axial plane, for evaluation and classification of the skeletal maturation stage of the MPS according to the method of Angelieri et al. [39].

According to the previous recommendations [39,40,50], in patients that had a curved palate, 2 axial sections were made: 1 section was in the front, and the other at the rear of the palate. This was done because of the impossibility of viewing the MPS in 1 axial section.

The study had 3 examiners: 1 radiologist (I.G.C.), 1 orthodontist (M.M.G.) and 1 general dentist (A.S.).

One experienced radiologist (I.G.C.) assessed all images and selected the best axial image according to the method of Angelieri et al. [40] (Figure 3). Subsequently, these images were saved as JPEG files and arranged sequentially in a presentation by the principal investigator (P.S.V.) (PowerPoint for Mac 2008; Microsoft, Redmond, Wash). The images were identified only by numbers. No change in contrast or brightness of these images was undertaken. Each patient was classified by the radiologist (I.G.C.), who was blinded, using a computer with a high-definition display in a dark room. This evaluation was considered the ground truth (the term “ground truth” is more frequently used regarding a consensus of radiographic interpretations or reliable interpretations). Consensus among radiographic interpretations or more reliable interpretations should not be considered a gold standard, because a gold standard would require histologic or microcomputed tomography examination of specimens.

### 2.1. Training and Calibration Process

The training was divided into two phases: a theoretical phase and a practical phase. Both of these were conducted by an experienced and trained radiologist (I.G.C.).

For the training session, all the examiners received training material that consisted of (1) an explanation of the method with detailed descriptions of the maturation stages; (2) a diagram retrieved from the original study of Angelieri et al. [40] used to support the classification; (3) a table designed from the diagram; and (4) CT images provided with their respective schematic drawings also retrieved from the original study of Angelieri et al. [40]; and an axial slice as an example of each stage.

The examiners met a week later to clarify any doubts that existed regarding the method as a whole and analyzed 10 images in which different stages of maturation could be identified. Any disagreements were discussed to obtain a final decision.

Subsequently, 20 different images were used to perform the calibration process. This analysis was performed twice by each examiner with a washout period of 4 weeks. On the second occasion, images were presented in another random order. Recorded data were submitted to the agreement analysis to check inter-examiner errors. In addition, the observer was asked to re-observe the slices to rule out any intra-examiner error. The weighted kappa coefficient was used for both analyses. These images were not included in the final sample.

The 116 CBCT sagittal slices were ordered in a PowerPoint presentation with a black background and codes that were displayed sequentially on a high-definition computer monitor. The level of midpalatal suture maturation was classified by each examiner (M.M.G. and A.S.) and tabulated in a research sheet designed by the principal investigator (P.S.V.). Four weeks after the first observation, both examiners received the images in a different random order and were asked to re-observe the slices. It is important to mention that both examiners were blinded to the results obtained by the radiologist.

### 2.2. Statistical Analyses

The weighted kappa coefficients were calculated for evaluation of the intra- and inter-examiner measurement error using SPSS IBM version 23 (SPSS Inc., Armonk, NY, USA), and the results were interpreted according to the scale of Landis and Koch [51]. All statistical procedures were conducted with SPSS IBM version 23 (SPSS Inc., Armonk, NY, USA) software for Windows.

Data collection was recorded in a Microsoft Office Excel spreadsheet. Five tests were performed by a biostatistician. To verify the normality of the data, the Kolmogorov–Smirnov test was applied. A U Mann–Whitney test was performed for independent samples. The chi-square test was used to analyze the possibility of finding an open midpalatal suture by age groups. Chronological age was compared among the maturation stages of the suture using the Kruskal–Wallis test. A binary logistic regression model was performed using the maturation stage of the MPS as an outcome variable. The predictor variables were age (in years) and sex (the codes were 0 and 1 for females and males, respectively). The impact of each factor on the outcome variable was expressed as an OR with its 95% confidence interval (95% CI). Statistical significance for all statistical tests was set at *p* < 0.05.

## 3. Results

In the calibration process, the weighted kappa coefficients for the evaluation of the intra-examiner measurement error in the MPS maturation stage were 0.89 for the orthodontist and 0.85 for the general dentist. In the same way, weighted kappa coefficients for the evaluation of the inter-examiner measurement error in the MPS maturation stage in the calibration process were 0.87 for the orthodontist and 0.81 for the general dentist.

In the final analysis of the samples, the weighted kappa coefficients for the evaluation of the intra-examiner measurement error in the MPS maturation stage were 0.87 for the orthodontist and 0.83 for the general dentist. In the same way, weighted kappa coefficients for the evaluation of the inter-examiner measurement error in the MPS maturation stage in the calibration process were 0.86 for the orthodontist and 0.81 for the general dentist. These values demonstrate almost perfect agreement according to the scale of Landis and Koch [51].

The most frequent maturation stage in the study population (Table 1) was Stage D (37.8%), followed by Stage C (24%), Stage E (19.6%), Stage B (14.4%) and Stage A (3.3%). The MPS was not fused in 49 out of 116 subjects (39.6% of the total sample with Stages A, B and C). In females, there was a higher prevalence of Stage D (10.3%) in the age group of 10 to 15 years.

In females, there was a higher prevalence of Stage D (10.3%) in the age group of 10 to 15 years. A similar situation happened with male subjects of this age group, in which there was a higher prevalence of stage D (9.4%), followed by nine subjects (7.7%) in stage C (Table 2).

In the group of 16- to 20-year-olds, males had a higher prevalence of Stage C, being present in six subjects. In females, the highest prevalence seen was Stage E (five patients) (Table 3).

In the group between 21 and 25 years old, no subject was observed in stage A. Nevertheless, stage B and stage C were found in six and seven subjects, respectively (Table 4).

When considering all the subjects (Table 5), we observed that in the age group of 10 to 15 years, it is possible to verify open MPS in 41.5%, and in the age group of 16 to 20 years and 21 to 25 years, it is possible to verify a MPS opening in 48.2% and 38.2%, respectively. Moreover, the findings show that in the three age groups, males have a higher possibility of presenting a MPS opening than females (61.2% vs. 38.8%).

The comparison of the maturation stages by sex is given in Table 6. Both sexes had a higher prevalence of stage D, which is more frequent in females (37.7%), followed by stage E (31.1%). Stage C was observed in 16.3%. The prevalence of stage A and B in females was 1.6% and 13.1% of the total sample. In males, Stage D was the most prevalent (38.1%), followed by Stage C (32.7%). Stage A was observed in 5.4%, stage B in 16.3% and stage E in 7.2%.

The results of the logistic regression (Table 7) showed that women are 2653 times more likely to have a closed midpalatal suture than men.

## 4. Discussion

The treatment of transverse maxillary constriction in patients is an important topic for orthodontists, and this is especially challenging in late-stage adolescent and young adult patients [48].

Despite the unquestionable success of the RME protocol in clinical practice, there is still no consensus regarding the age limit for conventional palatal expansion. This is mainly due to the great physiologic variability among patients in relation to the obliteration of palatal suture earlier or at a more advanced age, making it difficult to have a precise diagnostic [33,34,36,37,44]. This is important, because the greater the obliteration of the MPS, the lower the skeletal effects and the greater the dentoalveolar impact of this treatment [50,52].

This uncertainty creates insecurity. Even though RME is a more conservative treatment, if it is indicated in patients with totally or partially closed midpalatal sutures, it can lead to consequences, such as significant pain, gingival recession, palatal mucosa ulceration or necrosis, buccal tipping of the posterior teeth, reduction in buccal bone thickness [13,13,14,15,16,17,18], alveolar bone bending [53], buccal root resorption [54], fenestration of the buccal cortex [55] and instability of the expansion [56,57]. On the other hand, it is important to mention that even though a surgical expansion with SARPE is possible at any time throughout life, it implies increasing morbidity, cost, risk and more days required for patient recovery [58]. It has also been reported to be the most unpredictable procedure among all orthognathic surgery modalities. This unpredictability of the surgical expansion has to do with its relapse potential [59,60].

Due to the abovementioned issues, the treatment choice of whether an adolescent or a young adult patient is a suitable candidate for maxillary expansion without surgical assistance is a relevant clinical question [52].

CBCT imaging is a supplementary exam that can enable the clinician to three-dimensionally visualize the maxillary anatomy and evaluate the MPS maturation without the overlap of the surrounding structures such as the vomer or the nose on the MPS region, as happens in two-dimension occlusal radiographs [37].

The individual evaluation of midpalatal suture maturation in CBCT scans has been proposed by Angelieri et al., in order to identify the morphology of the midpalatal suture prior to intervention with one of the treatment options discussed previously for adolescent and young adult patients [40]. These researchers have reported great variability in the distribution of the maturational stages of midpalatal suture according to chronological age, with the fusion of the midpalatal suture in some female subjects older than 11 years and in some male subjects older than 14 years [40].

This method was designed to allow for a reliable and reproducible diagnostic pathway for clinical use [47].

To evaluate the morphology of the MPS according to the stages of maturation of Angieleri et al. [40], we measured CBCTs through a cross section in the middle of the palate.

Stages A to C indicate an open MPS, which is more suitable for a conventional RME approach, whereas Stages D and E are possibly related to suture closure, and a surgical approach could be preferred. According to the authors of [40], the assessment of MPS maturation can avoid the side effects derived from rapid maxillary expansion failure and limit surgically assisted rapid maxillary expansion to late adolescents and young adults with complete closure of the MPS, thus avoiding unnecessary treatment [61].

In both male and female patients, Stage D was most frequently observed (38.1% and 37.7%, respectively). However, Stage E was almost five times more prevalent in females than in male subjects. These findings may suggest that female patients have an advanced maturational status compared with male patients. This statement is confirmed by our study, because a statistically significant difference was observed between sexes (*p* = 0.012) (Table 7).

Stage A was found in only four subjects (3.4%), drawing attention to a 17-year-old male patient (Table 1). We expected that this stage would be observed in younger subjects, if it occurred at all. Likewise, Stage B had a low prevalence, found in 14.6% of the total sample. It is noteworthy that six of these patients belong to the group of 21 to 25 years. On the other hand, the prevalence of stages A and B in those over 18 years of age was only 6.8% in our study, 2.5% in the study of Ladewig et al. [39] and 3.1% in the study of Angelieri et al. [40].

Stage C is known to be a transitional stage. Tonello et al. mention that it is still favorable for obtaining maxillary expansion successfully. In our study, it was more present in male (32.7%) than female (16.3%) subjects, even being present in seven subjects of the group of 21- to 25-year-olds. Both Ladewig et al. [39] and Tonello et al. [50] stated that Stage C was the most prevalent: 44.6% and 50%, respectively. However, in our research, this stage was observed in 21.4% (Table 1).

Stages D and E were observed in our sample in 37.8% and 19.6%, respectively (Table 1). Jimenez et al. [48] reported a prevalence of 29% and 39% for Stages D and E. This difference could be the result of environmental and genetic characteristics of the sample evaluated [20,21,22,48].

Twelve females and eleven males of the age group of 10- to 15-year-olds were in Stage D (Table 2). In the same vein, eight females of the same age group were in Stage E. This was an unexpected result, since these stages represent the beginning or completion of the ossification of the suture, making it difficult for RME success.

In the study of Ladewig et al. [39], 60% of the subjects over 18 years of age were in Stage D and E. This value was even higher in the study of Angelieri et al. [40], in which 84.4% of the subjects were diagnosed in those stages. In our study, this value was significantly lower at 25.8%. A reason that could justify this difference is that in Angelieri’s study, the sample over 18 years was composed of subjects up to 59 years old, whereas in our study, the maximum age was 25 years.

In the group of patients between 21 and 25 years old, 38.2% still had their MPS partially or totally open (Table 5). Villarroel et al. [62] and Reis et al. [42] had similar values in the same age group, with 39.2% and 34.6%, respectively. This is important, because theoretically, conventional maxillary expansion could still be performed. It is important to mention that even though the MPS is still open in these patients, other craniofacial structures could offer resistance to palatal expansion. In fact, other circunmaxillary sutures [63], zygomatic arch [8,54] and sphenoid bone [8] are also involved. Therefore, in order to have a better prediction of the prognosis, these structures should also be taken into consideration in future studies [42].

Formerly, chronological age had traditionally guided the clinical decision between traditional RME and SARME, but it has been proven in literature that this is an unreliable indicator for determining the maturation status of MPS [1,6,12,48]. This assertion can be supported by the present study, leading to a paradigm shift in the treatment of maxillary constriction.

According to our study, the possibility of finding midpalatal suture opening is greater in the male sex compared to the female sex.

One of the strengths of this study was that examiners with different levels of experience and familiarity with CBCT in practice were selected in the present study to represent the clinical routine, where the method must be reproducible independently of the clinician. After an extensive training and calibration process, the values of kappa obtained for inter-examiner agreement and intra-examiner agreement demonstrated almost perfect agreement.

The clinical application of the method by orthodontists and general dentists would be questionable because of the amount of training and familiarity needed with the method itself. It is recommended that a report from a radiologist is requested [47]. On the other hand, it has a very big potential for research and educational purposes.

Other important strengths of this research are the image randomization process by an investigator that did not participate in the analysis of the samples, blinding of the observers and a sample size calculation before the execution of the study. We consider that it is very important to specify that the images used in the calibration process were not used in the main study. Further, images should be reorganized and randomized in a different order when assessing intra-examiner agreement.

Even though CT/CBCT technology allows clinicians and researchers to make a quantitative evaluation for the bone changes in three dimensions, it is important to consider that radiological assessment is not a risk-free procedure, especially when children are involved, and there is a growing concern of radiation dose in orthodontic CBCT [6,64,65,66].

This is the reason for which, if it is possible to establish the ossification status of the MPS of an individual with other biological indicators, it should be preferred to reduce the load of ionizing radiation to which the patient is exposed.

This study evidences the importance of individually analyzing the maturation of the MPS in each patient prior to maxillary expansion, avoiding potential adverse effects of each technique when not indicated properly. The possibility of children presenting partial or total maturation of MPS or young adults presenting earlier stages of maturation of the midpalatal suture should not be discarded.

In this context, the present study has allowed us to obtain valuable information about the MPS stages found in Chilean children, adolescents and young adults based on an extensive and homogeneous sample. This information could be a contribution to clinical decision making.

In the future, it would be recommended to perform clinical studies in which variations in the thickness of the MPS are considered, and an evaluation of circum-maxillary sutures is included.

## 5. Conclusions

-RME is a nonsurgical option that is clinically successful in most adolescents and young adults with a prevalence of stages A, B and C, because MPS is not fused. In our study, 42.2% of the sample was in those stages.-The majority of the patients (57.7%) presented partial or total fusion of the midpalatal suture (stages D and E).-Individual evaluation of each patient is crucial.-Assessment of midpalatal suture maturation on CBCT images, informed by a radiologist, may provide helpful information for the clinical decision between conventional or surgical expansion for the treatment of maxillary constriction.-This method has a very big potential for research and educational purposes.

## Figures and Tables

**Figure 1 diagnostics-13-01449-f001:**
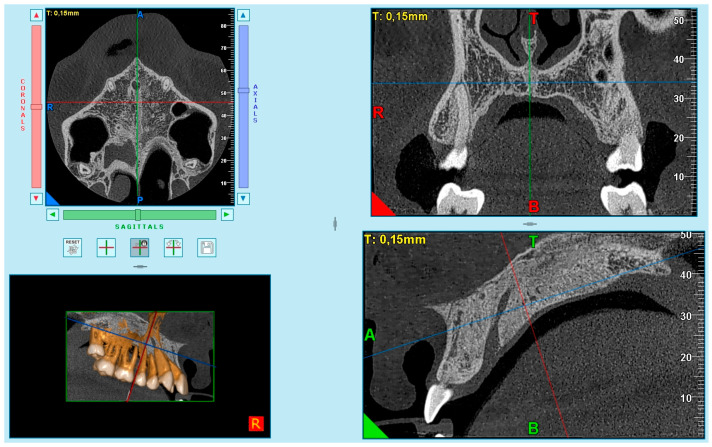
Anterior axial cut—palatal processes of the maxillary bone.

**Figure 2 diagnostics-13-01449-f002:**
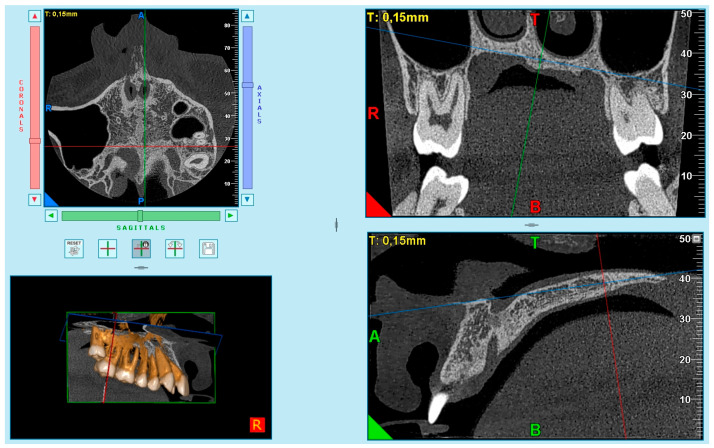
Posterior axial section—palatine processes of the maxillary bone.

**Figure 3 diagnostics-13-01449-f003:**
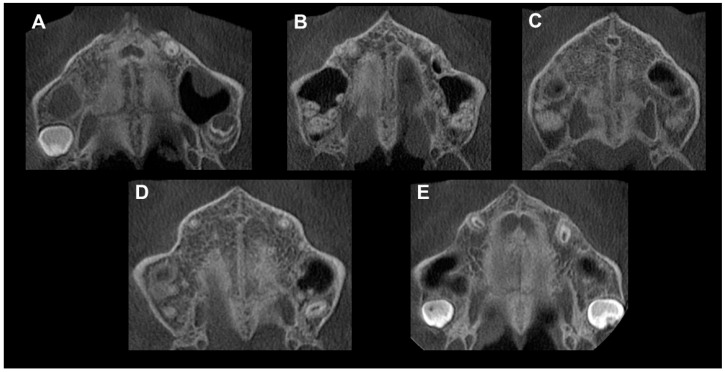
Method of Angelieri et al. [40] in CBCT. (**A**) The midpalatal suture is seen as a relatively straight radiopaque line. (**B**) The midpalatal suture appears as a scalloped line of high density. (**C**) Two radiopaque, scalloped and parallel lines are separated by areas of low radiographic density. (**D**) The palatine bones become more radiopaque, and the suture is not visualized in this sector; it is only visualized as two scalloped high-density lines at the midline on the palate bone. (**E**) It is no longer possible to see the suture along the maxillary and palatine bones, which indicates that fusion has occurred in the maxilla.

**Table 1 diagnostics-13-01449-t001:** Distribution of the MPS maturational stages by age group and sex.

Age (Years)	Sex	Stage										Total
		A		B		C		D		E		
		n	%	n	%	n	%	n	%	n	%	
10–15	F	1	0.8	5	4.3	2	1.7	12	10.3	8	6.8	28
	M	2	1.7	3	2.5	9	7.7	11	9.4	0	0	25
	F + M	3	2.5	8	6.8	11	9.4	23	19.8	8	6.8	53
16–20	F	0	0	2	1.7	4	3.4	4	3.4	5	4.3	15
	M	1	0.8	1	0.8	6	5.1	5	4.3	1	0.8	14
	F + M	1	0.8	3	2.5	10	8.6	9	7.7	6	5.1	29
21–25	F	0	0	1	0.8	4	3.4	7	6.0	6	5.1	18
	M	0	0	5	4.3	3	2.5	5	4.3	3	2.5	16
	F + M	0	0	6	5.1	7	6.0	12	10.3	9	7.7	34
TOTAL		4	3.3	17	14.4	28	24	44	37.8	23	19.6	116
F.: female, M.: male												

**Table 2 diagnostics-13-01449-t002:** Distribution of the MPS maturational stages by age and sex in 10- to 25-year-olds.

Age (Years)	Sex	Stage										Total
		A		B		C		D		E		
		n	%	n	%	n	%	n	%	n	%	
10	F	0	0	1	1.1	0	0	2	3.7	0	0	5
	M	0	0	0	0	3	5.6	1	1.8	0	0	4
	F + M	0	0	1	0	3	5.6	3	5.6	0	0	7
11	F	0	0	1	1.8	0	0	5	9.4	1	1.8	7
	M	0	0	0	0	0	0	2	3.7	0	0	2
	F + M	0	0	1	1.8	0	0	7	13.2	1	1.8	9
12	F	0	0	3	5.6	2	3.7	2	3.7	3	5.6	10
	M	1	1.8	1	1.8	0	0	4	7.5	0	0	6
	F + M	1	1.8	4	7.5	2	3.7	6	11.3	3	5.6	16
13	F	1	1.8	0	0	0	0	0	0	1	1.8	2
	M	1	1.8	2	3.7	2	3.7	1	1.8	0	0	6
	F + M	2	3.7	2	3.7	2	3.7	1	1.8	1	1.8	8
14	F	0	0	0	0	0	0	1	1.8	2	3.7	3
	M	0	0	0	0	2	3.7	3	5.6	0	0	5
	F + M	0	0	0	0	2	3.7	4	7.5	2	3.7	8
15	F	0	0	0	0	0	0	2	3.7	1	1.8	3
	M	0	0	0	0	2	3.7	0	0	0	0	2
	F + M	0	0	0	0	2	3.7	2	3.7	1	1.8	5
F.: female, M.: male												53

**Table 3 diagnostics-13-01449-t003:** Distribution of the MPS maturational stages by age and sex in group of 16- to 20-year-olds.

Age (Years)	Sex	Stage										Total
		A		B		C		D		E		
		n	%	n	%	n	%	n	%	n	%	
16	F	0	0	0	0	2	6.8	0	0	0	0	2
	M	0	0	0	0	0	0	0	0	0	0	0
	F + M	0	0	0	0	2	6.8	0	0	0	0	2
17	F	0	0	1	3.4	1	3.4	1	3.4	2	6.8	5
	M	1	3.4	0	0	3	10.3	2	6.8	1	3.4	7
	F + M	1	3.4	1	3.4	4	13.7	3	10.3	3	10.3	12
18	F	0	0	0	0	0	0	1	3.4	0	0	1
	M	0	0	0	0	2	6.8	1	3.4	0	0	3
	F + M	0	0	0	0	2	6.8	2	6.8	0	0	4
19	F	0	0	0	0	0	0	1	3.4	3	10.3	4
	M	0	0	0	0	0	0	0	0	0	0	0
	F + M	0	0	0	0	0	0	1	3.4	3	10.3	4
20	F	0	0	1	3.4	1	3.4	1	3.4	0	0	3
	M	0	0	1	3.4	1	3.4	2	6.8	0	0	4
	F + M	0	0	2	6.8	2	6.8	3	10.3	0	0	7
F.: female, M.: male												29

**Table 4 diagnostics-13-01449-t004:** Distribution of the MPS maturational stages by age and sex in group of 21- to 25-year-olds.

Age (Years)	Sex	Stage										Total
		A		B		C		D		E		
		n	%	n	%	n	%	n	%	n	%	
21	F	0	0	0	0	0	0	2	5.8	0	0	2
	M	0	0	3	8.8	0	0	1	2.9	0	0	4
	F + M	0	0	3	8.8	0	0	3	8.8	0	0	6
22	F	0	0	0	0	0	0	1	2.9	2	5.8	3
	M	0	0	2	5.8	1	2.9	0	0	2	5.8	5
	F + M	0	0	2	5.8	1	2.9	1	2.9	4	13.7	8
23	F	0	0	0	0	1	2.9	1	2.9	0	0	2
	M	0	0	0	0	0	0	1	2.9	0	0	1
	F + M	0	0	0	0	1	2.9	2	5.8	0	0	3
24	F	0	0	1	2.9	1	2.9	2	5.8	1	2.9	5
	M	0	0	0	0	1	2.9	2	5.8	0	0	3
	F + M	0	0	1	2.9	2	5.8	4	13.7	1	2.9	8
25	F	0	0	0	0	2	5.8	1	2.9	3	8.8	6
	M	0	0	0	0	1	2.9	1	2.9	1	2.9	3
	F + M	0	0	0	0	3	8.8	2	5.8	4	13.7	9
F.: female, M.: male												34

**Table 5 diagnostics-13-01449-t005:** Distribution of MPS maturational stages by age group and sex regarding the possibility of finding midpalatal suture opening.

Midpalatal Suture Opening						
Possibility				No Possibility		
Age (Years)	Sex	n	%	n	%	Total
10–15	F	8	6.8	20	17.2	28
	M	14	12	11	9.4	25
	F + M	22	18.9	31	26.7	53
16–20	F	6	5.1	9	7.7	15
	M	8	6.8	6	5.1	14
	F + M	14	12	15	12.9	29
21–25	F	5	4.3	13	11.2	18
	M	8	6.8	8	6.8	16
	F + M	13	11.2	21	18.1	34

F.: female, M.: male.

**Table 6 diagnostics-13-01449-t006:** Distribution and comparison of the MPS maturational stages in 10- to 25-year-old subjects by sex.

Age (Years)	Stage										Total
	A		B		C		D		E		
	n	%	n	%	n	%	n	%	n	%	
Female	1	0.8	8	6.8	10	8.6	23	19.8	19	16.3	61
Male	3	2.5	9	7.7	18	15.5	21	18.1	4	3.4	55

F.: female, M.: male.

**Table 7 diagnostics-13-01449-t007:** Results of the logistic regression model with the maturational stages of the midpalatal suture as outcome variable and age and sex as predictors.

Variable	*p*	OR
Sex	0.012 *	2.653
Age	0.699	1.015

* Statistically significant difference.

## Data Availability

https://doi.org/10.17605/OSF.IO/HW6BM (accessed on 1 February 2023).

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
