# Peer review of "Midpalatal Suture Maturation Stage in 10- to 25-Year-Olds Using Cone-Beam Computed Tomography—A Cross-Sectional Study"

_diagnostics, 2023, doi:10.3390/diagnostics13081449_

Round 1
Reviewer 1 Report
Dear Authors,
you made a great work! However, some improvements are mandatory before acceptance.

Author Response
February 2nd, 2023
RESPONSE TO REVIEWER Nº1
Dear Reviewer,
We would like to start by thanking you for the generous words and time that you have dedicated to our work.
In the following paragraphs, we will mention the changes made based on your requirements:
- We have asked two native English speakers to check the spelling and grammar used in our article.
- In the introduction, we have added (all written in red) and cited (citation number 45) the article written by Perrotti et al. We would like to thank you for this suggestion, since it is an article very relevant to what we are trying to express.
“Having a 3D diagnostic exam available is useful for the clinician when deciding the best treatment strategy for each patient, avoiding the various radiographic investigations of-ten necessary to obtain all the essential information needed to draw up the orthodontic treatment plan [45]”.
- We have checked all the text and corrected the double spaces.
We are at your disposal in case there is any other correction that you consider necessary.
Best regards,
The authors.

Reviewer 2 Report
Dear Authors,
please consider my evaluations with the hope of adding something to the manuscript.

Author Response
February 2nd, 2023
RESPONSE TO REVIEWER Nº2
Dear Reviewer,
We would like to start by thanking you for the generous words and time that you have dedicated to our work.
In the following paragraphs, we will mention the changes made based on your requirements:
- Please indicate in detail in the introduction if there is an alternative to the administration of this radiation dose with CBCT with less invasive methods for the patient subjected to ionizing radiation.
In relation to what was requested, we have added the following paragraphs (highlighted in red in the text):
In orthodontics, several methods have been described for the evaluation of the skeletal age [25] including hand wrist radiography [26, 27] and cervical vertebral maturation (CVM) based on lateral cephalogram [28-31].
Radiation exposure to pediatric patients who are more radiosensitive than adults should be as low as possible. In this sense, it is very important to use reliable indicators such as hand-wrist radiographs with effective dose 0.16 μSv[32] for predicting the midpalatal suture maturation.
In the case of the assessment of the skeletal age with the CVM, it is done on a cephalo-metric radiograph, routinely used in orthodontic practice, which makes it easy to apply. The use of an X-ray protective thyroid collar is of crucial importance. The effect dose of a lateral cephalogram without a thyroid collar was higher than a hand-wrist radiograph as 5.03 μSv[32].
Even though both methods have been proven very useful for the estimation of skeletal age, they don’t allow clinicians to observe the midpalatal suture in situ.
An important point to have in consideration is that it is impossible to regularly take CBCT from every pediatric patient, with an effective dose ranging between 19 and 368 μSv[32], due to ethical concerns about unnecessary radiation exposure. For this reason, each pa-tient must be evaluated individually and in detail.
- Please indicate in the manuscript what are the reasons why it was necessary to submit the patient to a CBCT, in more detail.
In relation to what was requested, we have added the following paragraphs (highlighted in red in the text):
The primary justification for the CBCT request was the diagnosis of retained teeth (such as canine impaction), skeletal malocclusion, assessment of third molars, evaluation of dis-crepancy between maxilla and mandible, dental inclination or thickness of bone tables.
We are at your disposal in case there is any other correction that you consider necessary.
Best regards,
The authors.

Reviewer 3 Report
The article is suitable for pubblication and can confirm something already present in literature, leading the clinical choices. Please add only the different reasons for wide FOV cbct in really young group of age.
Author Response
February 2nd, 2023
RESPONSE TO REVIEWER Nº3
Dear Reviewer,
We would like to start by thanking you for the generous words and time that you have dedicated to our work.
In the following paragraphs, we will mention the changes made based on your requirements:
Please add only the different reasons for wide FOV cbct in really young group of age.
Between the reasons for wide FOV used in the CBCTs obtained from young children we have added to the text:
The primary justification for the CBCT request was the diagnosis of retained teeth (such as canine impaction), skeletal malocclusion, assessment of third molars, evaluation of discrepancy between maxilla and mandible, dental inclination or thickness of bone tables.
We are at your disposal in case there is any other correction that you consider necessary. Thank you again for your kind words.
Best regards,
The authors.
